# Activated Fibroblast Program Orchestrates Tumor Initiation and Progression; Molecular Mechanisms and the Associated Therapeutic Strategies

**DOI:** 10.3390/ijms20092256

**Published:** 2019-05-07

**Authors:** Go J. Yoshida, Arata Azuma, Yukiko Miura, Akira Orimo

**Affiliations:** 1Department of Molecular Pathogenesis, Juntendo University Faculty of Medicine, 2-1-1 Hongo, Bunkyo-ku, Tokyo, 113-8421, Japan; 2Department of Pulmonary Medicine and Oncology, Graduate School of Medicine, Nippon Medical School, 1-1-5, Sendagi, Bunkyo-ku, Tokyo 1138603, Japan; s7081@nms.ac.jp

**Keywords:** angiogenesis, cancer-associated fibroblasts, extracellular matrix, fibrosis, heterogeneity, interstitial fluid pressure, metabolic reprogramming, transforming growth factor-β, tumor stiffness

## Abstract

Neoplastic epithelial cells coexist in carcinomas with various non-neoplastic stromal cells, together creating the tumor microenvironment. There is a growing interest in the cross-talk between tumor cells and stromal fibroblasts referred to as carcinoma-associated fibroblasts (CAFs), which are frequently present in human carcinomas. CAF populations extracted from different human carcinomas have been shown to possess the ability to influence the hallmarks of cancer. Indeed, several mechanisms underlying CAF-promoted tumorigenesis are elucidated. Activated fibroblasts in CAFs are characterized as alpha-smooth muscle actin-positive myofibroblasts and actin-negative fibroblasts, both of which are competent to support tumor growth and progression. There are, however, heterogeneous CAF populations presumably due to the diverse sources of their progenitors in the tumor-associated stroma. Thus, molecular markers allowing identification of bona fide CAF populations with tumor-promoting traits remain under investigation. CAFs and myofibroblasts in wound healing and fibrosis share biological properties and support epithelial cell growth, not only by remodeling the extracellular matrix, but also by producing numerous growth factors and inflammatory cytokines. Notably, accumulating evidence strongly suggests that anti-fibrosis agents suppress tumor development and progression. In this review, we highlight important tumor-promoting roles of CAFs based on their analogies with wound-derived myofibroblasts and discuss the potential therapeutic strategy targeting CAFs.

## 1. Significant Roles of Fibrosis in Cancer Development

### 1.1. Contributions of Fibrosis to Cancer Development

Injured epithelial tissues are repaired by the formation of granulation tissues rich in α-smooth muscle actin (α-SMA)-positive myofibroblasts (a hallmark of activated fibroblasts), platelets, newly formed blood vessels, macrophages, and other inflammatory cells and extracellular matrix (ECM). The transforming growth factor-β (TGF-β) signal pathway is involved in the emergence of myofibroblasts, which contribute to the production of matrix metalloproteinase (MMP) and ECM proteins, such as collagen I, fibronectin and hyaluronic acid [1,2,3,4]. The damaged tissues are then degraded and ECM proteins are simultaneously generated *de novo* [2,3,4,5]. Sustained activation of myofibroblasts promotes dysfunctional repair mechanisms, leading to accumulation of fibrotic ECM which is rich in collagen fibers and resistant to MMP-mediated degradation [1,6,7]. The fibrotic ECM inhibits epithelial cell polarity and stimulates epithelial cell proliferation, which in turn results in conditions allowing tumor formation and development [8,9].

In fact, a growing body of evidence suggests that the presence of fibrotic lesions significantly increases the risk of cancer in numerous tissues, including the lungs, liver and breast [8,9,10,11]. Idiopathic pulmonary fibrosis (IPF), which is a progressive and fatal lung disease of unknown etiology, is associated with a higher incidence of lung cancers as compared with the general population [12]. IPF is characterized by scar tissue accumulation in the lung interstitium. The injury to type II alveolar epithelial cells triggers production of TGF-β that leads to mitogenesis of macrophages, platelets and myofibroblasts in the injured areas, leading to the formation of fibroblastic foci. Fibroblastic foci containing myofibroblasts at the leading edge of lung fibrosis are an indicator of poor prognosis and decreased survival [13].

The secreted protein acidic and rich in cysteine (SPARC) family of proteins regulate ECM assembly and growth factor signaling to modulate interactions between cells and the extracellular environment [14,15]. SPARC (also known as osteonectin, an acidic extracellular matrix glycoprotein) binds to soluble procollagen and prevents procollagen from interacting with cellular receptors, such as discoidin domain receptor 2 and integrins [15,16]. In the absence of SPARC, procollagen accumulates at the cell surface and is inefficiently incorporated into the ECM, resulting in the production of thin collagen fibers. SPARC is thus required for procollagen to be dissociated from the cell surface and incorporated into the ECM.

SPARC is exclusively expressed in IPF patients, never in healthy individuals [9,17]. SPARC expression is also tightly correlated with increased collagen deposition. Inhibition of SPARC expression significantly attenuates fibrosis in various animal models of disease [15]. SPARC is also localized in the cytoplasm of the actively-migrating myofibroblasts within the fibroblastic foci [17]. SPARC expression and TGF-β signaling are reciprocally regulated; TGF-β induces SPARC expression via canonical Smad2/3 signaling in lung fibroblasts and SPARC which, in turn, activates TGF-β signaling [18]. TGF-β also induces plasminogen activator inhibitor-1 (PAI-1) expression via Smad2/3 signaling in lung fibroblasts. Moreover, SPARC-activated integrin promotes Akt activation that inhibits glycogen synthase kinase-3β (GSK-3β) by serine-9/21 phosphorylation, leading to β-catenin activation and PAI-1 expression [17]. As PAI-1 prevents lung fibroblasts from undergoing apoptosis induced by plasminogen, ectopic SPARC expression in IPF apparently mediates the progression of interstitial fibrosis by inhibiting apoptosis in lung myofibroblasts via β-catenin activation and PAI-1 expression in collaboration with the TGF-β signal pathway. Taken together, the observations of these cellular mechanisms by which SPARC promotes the activation of fibroblasts in culture and its fibrosis-promoting ability in vivo encourage investigators to seek therapeutic strategies for blocking SPARC activity. Such research may lead to the eradication of fibrotic diseases.

In contrast to the fibrosis-promoting SPARC function, the roles of stromal SPARC in human carcinomas appear to be far more complex and even contradictory according to previous reports. Enhanced SPARC expression in the tumor-associated stroma correlates with a poor prognosis for patients with non-small cell lung cancers (NSCLC) [19] and pancreatic adenocarcinomas [20], but not for those with bladder cancers [21]. Chemical agent-induced bladder carcinomas have been shown to grow and progress more significantly in SPARC^−/−^ mice than in control SPARC^+/+^ mice [21]. Murine carcinoma-associated fibroblasts (CAFs) extracted from SPARC^−/−^ bladder carcinomas also exhibit enhanced inflammatory phenotypes via NF-κB and AP-1 signaling, thereby promoting tumor growth and metastasis, indicating a tumor-suppressive role of SPARC in bladder CAFs. Collectively, these observations indicate cell-context dependent roles of stromal SPARC in different tumors.

Furthermore, non-alcoholic steatohepatitis (NASH), characterized by fat accumulation, inflammation and liver cell damage, leads to advanced fibrosis and cirrhosis, thereby increasing the risk of developing hepatocellular carcinoma (HCC) [22,23]. Diabetes mellitus (DM) with insulin resistance has also been demonstrated to be an independent risk factor for HCC development in NASH patients [23,24]. Activation of insulin-like growth factor 1 (IGF1) signaling stimulates cellular proliferation by activating the mitogen-activated protein kinase (MAPK) pathway and increases the transcription of c-Fos and c-Jun proto-oncogenes [25,26,27]. Moreover, phosphatase and tensin homologue deleted on chromosome 10 (PTEN) is a crucial negative regulator of the insulin signal pathway mediated by suppression of the phosphatidylinositol-3 kinase (PI3K)-Akt signal pathway. It has been shown that concomitant down-regulation of PTEN and up-regulation of c-Met occurs in HCC, leading to poor clinical outcomes [28]. Loss of PTEN function leads to the accumulation of phosphatidylinositol-3,4,5-triphosphate (PIP3), which mimics the effects of PI3K activation and triggers the activation of its downstream effectors, PDK1, Akt and Rac1/CDC42. Taken together, these observations demonstrate that NASH induces activation of an oncogenic signal transduction series of events in the non-cancerous liver to initiate tumor development.

### 1.2. Epithelial-Mesenchymal Transition (EMT) and Endothelial-Mesenchymal Transition (EndoMT) in Fibrosis and Tumor Stroma

As the saying “tumors: wounds that do not heal” goes, myofibroblasts in wounds and fibrosis mimic CAFs within a tumor [4,29,30] (Figure 1). Epithelial cells frequently transdifferentiate into mesenchymal cells through EMT during wound healing and fibrosis. EndoMT, another form of cellular transition, has also emerged as a mechanism underlying pathological fibrosis development [31,32,33]. Lineage-tagging experiments using a murine fibrosis model of renal injury indicate that about 30% of the cells involved are derived from tubular epithelial cells via EMT, while about 35% arise from EndoMT [34].

EndoMT is a complex biological process in which endothelial cells lose their molecular markers, such as vascular endothelial cadherin (VE cadherin), and acquire the myofibroblastic phenotype expressing mesenchymal markers including α-SMA, type I collagen, and vimentin. These cells also gain motility and are thus capable of migrating into the surrounding tissues [31]. TGF-β treatment induces the downstream signaling pathway to significantly upregulate Snail1 expression in endothelial cells via EndoMT [35,36,37]. This observation strongly suggests that EMT and EndoMT share the similar molecular mechanisms. Remarkably, TGF-β1-induced EndoMT occurs independently of Smad2/3 phosphorylation via non-canonical TGF-β signaling [31]. Furthermore, several important kinases including the c-Abl protein kinase (c-Abl), protein kinase C δ (PKC-δ) and GSK-3β, have been shown to play pivotal roles in this Smad-independent TGF-β pathway. In the absence of GSK-3β phosphorylation, the kinase activity of GSK-3β is promoted and induces proteasomal degradation of Snail1, thereby abrogating EndoMT. Both c-Abl and PKC-δ are required for GSK-3β phosphorylation to induce EndoMT [38]. From the perspective of preventing tissue fibrosis, the inhibition of GSK-3β serine-9 phosphorylation by a specific inhibitor of PKC-δ, i.e., rottlerin, or by c-Abl, widely known as imatinib, degrades Snail1 and thereby inhibits EndoMT [31]. Thus, rottlerin and imatinib both effectively suppress acquisition of the myofibroblastic phenotype and pathological fibrotic changes.

Myofibroblasts reportedly promote the induction and maintenance of EMT of epithelial cells at wound edges [4,39]. When this physiological EMT process is disrupted, wounds cannot heal. For instance, wound re-epithelialization of dermal tissue is compromised in mice lacking functional Slug, which is one of the transcription factors involved in TGF-β-induced EMT [40,41]. To achieve EMT at the wound edge, myofibroblasts secrete extracellular proteolytic enzymes, such as MMPs, which cleave ECM components and release potent TGF-β (latent form) and other EMT-inducing cytokines [6,42]. This is an intriguing parallel with the role EMT in cancer development and progression; while cancer cells are regulated by cytokines, such as TGF-β, these cytokines become tethered within the ECM, such that it remains ready for mobilization in response to certain triggers.

Stromal myofibroblasts are also observed in proximity to carcinoma cells associated with the EMT phenotype [43]. CAFs in this context participate in the release and bioavailability by secreting extracellular proteases and ECM-remodeling enzymes. It was previously shown that normal colonic fibroblasts differentiate into α-SMA-positive CAFs and secrete larger amounts of MMP2 and urokinase-type plasminogen activator (uPA) associated with various cancer cells [44,45]. These proteolytic enzymes have been suggested to cleave various ECM components such as decorin, which covalently and potently binds to TGF-β and prevents the potential ligand from binding to the TGF-β receptor in adjacent cancer cells [46]. ECM components act as a reservoir for various cytokines; since decorin is able to bind TGF-β1, proteolytic degradation of decorin results in the release of this sequestered TGF-β ligand [47]. These lines of evidence all strongly suggest that paracrine signaling from CAFs and myofibroblasts in a wound regulates epithelial-mesenchymal plasticity in nearby epithelial cells to further promote tumor progression and fibrosis, respectively.

## 2. Fibrosis-Induced Tumor Progression

### 2.1. Origin and Differentiation into CAFs

Although CAFs represent a major cellular component of the tumor stroma, a precise molecular definition of CAFs is as yet lacking. Attempts to define CAFs are usually aimed at identifying morphological features and expression patterns of the following proteins: α-SMA, asporin, collagen 11-α1 (COL11A1), fibroblast-activating protein (FAP), platelet-derived growth factor receptor (PDGFR) α/β, fibroblast-specific protein 1 (FSP1, also called S100A4), podoplanin, SPARC, S100A4, tenascin-C, microfibrillar-associated protein 5 (MFAP5), and vimentin [33,48,49,50]. However, none of these markers are specific to CAFs. Lack of the appropriate molecular markers for identifying tumor-promoting CAFs thus makes it difficult to elucidate the biology of these fibroblasts. Such a precise understanding would be the first, and most fundamental, step toward developing a cell type-specific targeting approach.

CAFs produce growth factors and inflammatory cytokines that are capable not only of regulating fibroblast activation in an autocrine fashion, but also of controlling the behaviors of cancer cells as well as other stromal cells, along with remodeling the ECM in a paracrine manner [33,51,52]. CAFs transdifferentiate from their progenitors, such as resident fibroblasts, endothelial cells, preadipocytes and bone marrow-derived mesenchymal stem cells (MSCs) during tumor progression [50,53,54,55]. MSCs are known to differentiate into CAFs in culture [56]. Injection of MSCs with carcinoma cells into immunodeficient mice also results in enhanced tumor growth and metastasis presumably through differentiation into tumor-promoting CAFs [57]. However, how differences among cells of origin for CAFs impact their biological functions has yet to be elucidated. A recent elegant study demonstrated the unique roles of CAFs originating from bone marrow in breast carcinomas [58]. Using MMTV-PyMT transgenic mice and adaptive bone marrow transplantation techniques, Raz et al. showed bone marrow-derived CAFs extracted from breast tumors to have proangiogenic traits and that, when implanted into recipient mice, these traits were significantly more marked than in those locally arising from the mammary gland. They also found decreased PDGFRα expression to allow bone marrow-derived CAFs to be distinguished from other CAF populations in breast tumors, highlighting distinct fibroblast populations present in the tumor.

Myofibroblastic CAFs are induced *de novo* from their progenitors including normal fibroblasts, when treated with TGF-β, platelet-derived growth factor (PDGF), Wnt7a, exosomes and microRNAs in culture [49,50,59,60,61]. Activation of fibroblasts with a pro-inflammatory state also occurs in otherwise non-activated fibroblasts treated with interleukin (IL)-1β, IL-6, leukemia inhibitory factor (LIF) and osteopontin [62,63]. However, it remains unknown whether these *de novo* generated CAFs continue to maintain their activated, tumor-promoting traits after a series of passages in culture or incubation with carcinoma cells within a tumor mass.

From the perspective of epigenetic modifications in myofibroblasts, LIF induces constitutive activation of the Janus kinase 1 (JAK1)/STAT3 signaling pathway mediated by post-translational regulation of STAT3 acetylation by p300 [64,65,66]. The acetylated STAT3 causes an epigenetic-dependent loss of expression of the Src homology region 2 domain-containing phosphatase-1 (SHP-1) tyrosine phosphatase, which is a negative regulator of the JAK/STAT pathway [67,68]. Silencing of SHP-1 gene expression by promoter methylation leads to sustained phosphorylation of JAK1 kinase and the STAT3 transcription factor that maintain the contractile and invasive abilities of CAFs [65]. Blockage of both JAK signal and DNA methyltransferase activities results, both in vitro and in vivo, in the reversal of the invasive phenotype of CAFs.

Table 1 shows the common activated signal pathways in both wound-induced activated fibroblasts and CAFs. This table comparatively details the biological roles of growth factors and cytokines in wound-healing and tumor stroma settings.

### 2.2. Emerging Roles of CAFs for Therapeutic Resistance

Recent emerging evidence supports crucial roles of CAFs for therapeutic resistance, as exemplified by innate and adaptive resistance in various human carcinomas (Figure 2).

#### 2.2.1. Innate Resistance of CAFs to Anti-Cancer Drugs

CAFs produce inherently increased levels of growth factors and inflammatory cytokines that attenuate the efficacy of anti-cancer treatment. For example, CAFs produce an abundance of insulin-like growth factor 2 (IGF2) that renders cholangiocarcinoma and pancreatic cancer cells resistant to EGFR tyrosine kinase inhibitors (TKI) by activating the insulin receptor (IR)/insulin-like growth factor 1 receptor (IGF1R) signaling axis [85,86]. CAF-produced IGF2 also reportedly promotes invasion and metastasis of colon cancer cells [87]. Moreover, stromal IGF2 induces NANOG expression and thus boosts the cancer-initiating properties of lung cancer cells through IR/IGF1R signaling followed by activation of the AKT-PI3K pathway [88].

Mechanisms underlying the stroma-mediated innate resistance to the BRAF inhibitor have also been addressed using BRAF-mutant melanoma cells in other studies. Hepatocyte growth factor (HGF) released from fibroblasts contributes to resistance to the BRAF inhibitor, presumably via the downstream signaling of the MAPK and PI3K/AKT pathways [89]. While treatment with BRAF- and MEK-inhibitors is not sufficient to overcome HGF-induced resistance, BRAF- and MET (the HGF receptor)-inhibitors suppress the majority of HGF-induced drug resistance in BRAF-mutant melanoma [89,90]. Similarly, HGF-producing human fibroblastic cells confer resistance to gefitinib, a TKI selective to EGFR in lung cancer cells through the PI3K/Akt signal pathway [91]. Importantly, anti-HGF neutralizing antibody and the natural HGF inhibitor NK4 significantly overcome the gefitinib resistance in culture and in tumor xenografts raised by lung cancer cells admixed with HGF-producing human fibroblasts in mice treated with gefitinib.

Inhibition of VEGF-A is effective in treating several human carcinomas [92]. However, tumors often show resistance to anti-VEGF treatment. Importantly, CAFs have been shown to mediate the resistance to anti-VEGF therapy and the molecular mechanism was elucidated in murine lymphoma models [93]. Murine CAFs were isolated from subcutaneous tumors developed by lymphoma cell lines resistant to the antiangiogenic therapy with VEGF inhibitors. The increased level of PDGF-CC produced by CAFs resulted in rendering tumor cells resistant to anti-VEGF therapy by stimulation of neoangiogenesis, when lymphoma cells otherwise sensitive to anti-VEGF therapy were co-implanted with these fibroblasts into recipient mice [93].

Human breast cancers of the luminal subtypes expressing female hormone receptors are effectively treated with selective estrogen receptor modulators (SERMs), such as tamoxifen, while tumors of the basal-like subtype that do not express hormone receptors fail to have effective targeted therapies. Roswall et al. have recently demonstrated that CAFs play crucial roles in regulating the phenotypic conversion of luminal breast cancers into basal-like cancers, which show the worst overall survival among various human breast cancer subtypes [77]. Human breast cancer cells produce PDGF-CC that acts onto the cognate PDGF receptors expressed on closely apposed CAFs to activate these fibroblasts. The resulting activated CAFs produce stanniocalcin 1 (STC1), HGF and insulin growth factor binding protein 3 (IGFBP3), all of which downregulate the expression levels of the luminal markers including FOXA1, estrogen receptor and GATA3, resulting in the conversion of the luminal tumors into basal-like tumors [77]. Notably, the luminal phenotype and sensitivity to endocrine therapy were shown to be restored in otherwise resistant tumors, not only by genetic targeting of the PDGF-C gene in the MMTV-PyMT murine basal-like breast cancer model, but also by treatment with the neutralizing PDGF-CC antibody of patient-derived triple-negative breast tumor xenografts transplanted orthotopically in immunodeficient mice.

#### 2.2.2. Adaptive Resistance of CAFs to Anti-Cancer Drugs

CAFs have been shown to prime chronic inflammation, as exemplified by recruitment of protumorigenic macrophages in an NFκB signal-dependent fashion, resulting in the promotion of tumor growth and angiogenesis [94]. With stress exposures, such as chemotherapy and radiation, these fibroblasts also acquire the pro-inflammatory phenotype via further activating NFκB signaling, resulting in increased survival signals of cancer cells. The activated NFκB signaling in the therapy-treated CAFs enhances production of different cytokines including Wnt family member wingless-type MMTV integration site family member 16B (WNT16B), IL-6 and IL-8, leading to the induction of chemoresistance in breast cancer cells [95,96]. CAFs treated with cisplatin also boost IL-11 production to activate the STAT3 signal pathway and upregulate the expressions of the anti-apoptotic proteins such as Bcl-2 and survivin for prostate cancer cells to acquire resistance to apoptosis [97]. Moreover, CAFs extracted from freshly resected human colorectal cancer specimens after chemotherapy reportedly show higher IL-7 production than those without chemotherapy [98]. This stromal IL-7 provides CD44-positive colon cancer cells with further increased tumor-initiating ability, thereby promoting tumor cell growth both in vitro and in vivo [98].

Furthermore, chemotherapy-induced stromal chronic inflammation is responsible for angiogenesis and ECM remodeling, and subsequently provides tumor cells with a physical barrier against the cytotoxic agents administered [33,99,100]. Recent investigations have shown that conventional chemotherapy and radiotherapy can lead to increased tumor stiffness involving the stroma response [49,101,102]. Remarkably, treatment with PLX-4270 (BRAF inhibitor) activates tumor-associated fibroblasts to induce ECM remodeling that activates integrin β1/FAK/Src signaling in melanoma cells [101]. This signal activation then renders melanoma cells resistant to PLX-4270 via ERK activation.

Treatment of stromal fibroblasts with high concentrations of HDAC inhibitors, such as SAHA, TSA and vorinostat, causes the senescence-associated secretory phenotype (SASP) mediated by the direct activation of the NFκB signal [103,104]. Treating fibroblasts with HDAC inhibitors results in significant paracrine stimulation of tumor growth, which suggests that high-dose HDAC inhibitors would likely impact the stromal compartment adversely in a therapeutic setting.

Intriguingly, androgen deprivation therapy increases the population of CD105 (endoglin)-positive CAFs, which contribute to neuroendocrine differentiation of epithelial prostate carcinoma cells [105]. CAF-derived secreted frizzled-related protein 1 (SFRP1), which is driven by the CD105-mediated signal pathway, is both necessary and sufficient to induce the neuroendocrine differentiation of prostate carcinoma in a paracrine manner. These series of observations raise concern regarding the undesirable side effects of therapeutic agents; paracrine signaling from treatment-primed CAFs might influence the regrowth and malignancy of nearby tumor cells.

CAFs play a key role in driving the drug resistance not only by raising particular gene expression patterns and signaling pathways as mentioned above, but also by stimulating recruitment of immunosuppressive cells into the tumor. Tumor-associated macrophages (TAMs) are non-neoplastic cells abundant in stroma of different human tumors and exert either pro-tumoral or tumoricidal functions in response to cytokine exposure [106,107]. A growing body of evidence indicates that pro-tumoral TAMs support tumor growth and progression by influencing tumor hallmarks [107]. Colony-stimulating factor 1 (CSF1) receptor signaling is a key regulator of TAM recruitment, differentiation and survival. Treatment with CSF1 receptor inhibitors targeting TAMs clearly reduces tumor growth in murine tumor models, though the anti-tumor effect was shown to be very limited in patients [106]. The molecular mechanisms underlying the tumor progression elicited by substantial depletion of TAMs remain, however, unknown. Importantly, a recent study revealed this to be due to increased CCL3 and CXCL-1/2/5 productions from CAFs treated with the CSF1 receptor inhibitor [108]. These CAF-produced chemokines then stimulate the recruitment of polymorpho-nuclear myeloid-derived suppressor cells (PMN-MDSCs) into tumors, resulting in the promotion of tumor growth and progression. These findings therefore demonstrate that CAFs mediate neutralization of the anti-tumor effect exerted by CSF1 receptor inhibitors via recruitment of PMN-MDSCs into tumors.

### 2.3. Cross-Talk between CAFs and Tumor Microenvironment

The wound-healing program is strongly dependent on the cross-talk between various stromal cells and myofibroblasts at the wound site [109,110] (Figure 1). For instance, myofibroblasts induce angiogenesis from preexisting parental vessels or from the circulating endothelial precursor cells (EPCs) recruited at the wound site via the secretion of a potent preangiogenic chemokine, CXCL12, also known as stromal cell-derived factor-1 (SDF-1) [111,112]. Chemotactically-attracted EPCs then transdifferentiate into endothelial cells in the presence of VEGF, which is also secreted by myofibroblasts.

In certain contexts, myofibroblastic CAFs induce neo-angiogenesis via secretion of preangiogenic factors including CXCL12, VEGF, PDGF, TGF-β and HGF in a wide range of cancers [49,50,113,114,115,116,117]. Most desmoplastic tumors are highly vascularized, wherein shifting the switch toward an angiogenesis-promoting phenotype occurs [118,119]. Interestingly, as in the case of wound healing, the CAF niches depend on the CXCL12/CXCR4 axis and VEGF production to stimulate the formation of neovasculature at the invasive front of breast cancer [115,120,121,122]. As the CXCR4 receptor for CXCL12 is expressed on both the tumor cells and EPCs [123,124], CAF-produced CXCL12 stimulates tumor growth and neoangiogenesis via acting CXCR4 expressed on these cells. The production of VEGF from tumor cells and CAFs also boosts neoangiogenesis in breast cancer tissues. Collectively, niches of myofibroblasts in wounds and CAFs are both likely to support angiogenesis, apparently through similar signaling pathways.

It was recently shown that hypermethylated in cancer 1 (HIC1), which is a tumor suppressor gene located at 17p13.3, resides exclusively within CpG islands, frequently showing hypermethylation in several tumors such as breast, lung and prostate carcinomas [125,126,127,128]. A recent study found that HIC1-depleted breast cancer cells markedly produce CXCL14 that activates Akt and ERK1/2 signal pathways, by acting through its cognate receptor GPR85 on resident fibroblasts in a paracrine manner, resulting in the induction of a phenotypic conversion into CAFs [127]. The activated CAFs, in turn, boost the production of chemokine CCL17 that acts on its cognate receptor CCR4, located on breast cancer cells, to drive metastasis. Collectively, the HIC1-CXCL14-CCL17 positive-feedback loop reciprocally mediating interactions between breast tumor cells and myofibroblastic CAFs contributes to the malignant potentials of breast tumors.

CAFs remodel the ECM components and thereby regulate tumor stiffness [129]. It was recently shown that squamous cancer cells activate EGFR in response to tumor stiffness, which leads to actomyosin contractility and collective invasion [130]. From a mechanistic standpoint, enhanced tyrosine kinase activity of EGFR results in Ca^2+^-dependent regulation of Cdc42 small GTPase activity in tumor cells, which in turn leads to phosphorylation of MLC2. The MLC kinase regulates actomyosin-dependent ECM remodeling due to CAFs. Surprisingly, two Ca^2+^ channel blockers, the phenylalkylamine verapamil and the nondihydropiridine diltiazem, which have been used for treating hypertension and arrhythmia for decades, show the therapeutic efficacy for preventing collective cancer invasion, an effect achieved by significantly down-regulating the phosphorylation of MLC2.

It is noteworthy that mechanical force-mediated ECM remodeling by CAFs depends on actomyosin contractility generated through the Rho-associated protein kinase (ROCK) signal pathway [129]. The IL-6/JAK1/signal transducer and activator of transcription 3 (STAT3) axis also controls actomyosin contractility by regulating the levels of phosphorylated-MLC2 in both melanoma cells and CAFs. In striking contrast to melanoma cells, in which the IL-6-gp130/ JAK1-ROCK axis is required for the amoeboid-like individual tumor migration, this signaling pathway is not required for the migration of squamous carcinoma cells themselves, but is required for CAFs to remodel the matrix, which is necessary for promoting the collective invasion of these carcinoma cells [64].

EMT of epithelial tumor cells, the process by which the number of tumor-initiating cells (TICs) is increased [131,132], is apparently regulated in cooperation with CAFs. Recent studies also highlight the importance of epithelial-mesenchymal plasticity to be determined by several transcription factors, including ZEB1, Snail and Twist. The resulting tumor cells with the hybrid epithelial/mesenchymal trait mediated by partial EMT are considered to enhance the tumor-initiating, invasive and metastatic properties as well, along with promoting chemoresistance [132,133,134,135]. These phenotypic changes are also presumably induced by CAF-regulated ECM components, exosomes and soluble factors.

## 3. Metabolic Reprogramming of CAFs During Cancer Progression

### 3.1. Metabolic Symbiosis Between Cancer Cells and CAFs

Features of desmoplastic tumor stroma resemble those of wound healing and involution during gestation. Mammography measures and compares the different types of breast tissue visible on a mammogram, which is an X-ray image of the breasts routinely used to screen for breast cancer in clinics. High breast density represents a greater amount of glandular and connective tissue than fat and has an association with higher risk for breast cancer development [136]. The expression level of CD36, a cell surface receptor for fatty acids, is downregulated in fibroblasts extracted from noncancerous breast tissues with high mammographic density as well as in breast CAFs [137]. CD36 expression is also known to be required for human mammary fibroblasts to transdifferentiate into preadipocytes in culture. Consistently, adipocytes were shown to regenerate from myofibroblasts in a murine skin wound healing model [138]. These findings demonstrate that down-regulation of CD36 expression in the stroma results in increased fibrosis in the breast, resulting in high mammographic density, presumably via attenuated transdifferentiation into adipocytes from mammary fibroblasts.

AMP-activated protein kinase (AMPK) reportedly regulates the translocation of the fatty acid transporter CD36 from intracellular stores to the plasma membrane [139], thereby promoting fatty acid uptake into skeletal muscle. CD36 also contributes to the activation of mitochondrial fatty acid β-oxidation (FAO) which in turn influences their metabolic plasticity in ovarian and oral carcinoma cells, leading to greater lymph node metastasis [140,141].

CD36 is involved in caveolae, a subset of lipid rafts forming part of the cell membrane microdomain enriched in cholesterol and signaling proteins. Decreased expression of CAV1, another component of caveolae within the tumor microenvironment, is also consistently associated with poor clinical outcomes in patients with a wide variety of malignancies [142]. CAV1-deficient fibroblasts also show concomitantly decreased CD36 expression, stabilization of hypoxia-induced factor-1α (HIF-1α), activation of TGF-β signal transduction and induction of myofibroblast differentiation [142,143]. These CD36-deficient fibroblasts likewise undergo a metabolic shift from mitochondrial oxidative phosphorylation to aerobic glycolysis, promoting the metabolic plasticity of these fibroblasts [142,144,145] (Figure 3). These findings suggest that altered caveolae function in the tumor microenvironment induces tumor metabolic heterogeneity, leading to the manifestation of malignant features. These mechanistic insights into how the alteration of caveolae is induced and maintained in CAFs are currently under investigation.

Initially, the Warburg effect was believed to be confined to specific tumor cell types [146,147,148]. However, the emerging concept of a “reverse Warburg effect” has recently attracted considerable research attention [145,149,150]. Tumor-derived reactive oxygen species (ROS) are responsible for down-regulation of CAV1 in CAFs [151,152,153] (Figure 3). Loss of CAV1 in CAFs also results in ROS elevations, which in turn stabilize HIF-1α. In other words, malignant cells induce a “pseudo-hypoxic” microenvironment for CAFs [145,154]. Because the transcription factor HIF-1α promotes glycolysis and provides cancer cells with lactate and glutamate, elevated ROS production in tumor cells indirectly induces the uptake of intermediate metabolites of the tricarboxylic acid (TCA) cycle in mitochondria (Figure 4). Of note, CAFs consume more glucose and secrete more lactate than normal fibroblasts [154,155,156]. Furthermore, CAFs depend significantly on autophagy that may lead to resistance to chemotherapy [145,155,157]. Collectively, fibroblasts surrounding epithelial tumor cells undergo metabolic reprogramming, which results in a metabolic phenotype resembling that induced by the Warburg effect. Importantly, metabolic symbiosis between epithelial cancer cells and CAFs requires a cell population to express a different MCT subtype [145,158,159,160,161]. Epithelial tumor cells express MCT1, which contributes to uptake of the lactate provided by CAV1-deficient CAFs, which in turn express MCT4, a marker of both aerobic glycolysis and lactate efflux (Figure 4).

### 3.2. Signal Pathways Involved in Metabolic Reprogramming of CAFs

Accumulating evidence strongly suggests that p62 (also known as sequestosome 1) is involved in metabolic reprogramming of activated fibroblasts in fibrosis and tumor stroma [162,163,164,165,166]. p62 is a multifunctional adaptor protein and a specific substrate for autophagy. p62 is thus selectively incorporated into autophagosomes through the direct binding with LC3 (microtubule-associated protein light chain 3) to be degraded by autophagy [167,168].

Hepatic stellate cells (HSCs), which can transdifferentiate into myofibroblasts in response to certain stimuli, play critical roles in liver fibrosis and HCC development [163]. A study showed that vitamin D receptor (VDR) signaling exerts the anti-fibrotic and anti-inflammatory effects in HSCs [163]. p62 also mediates the anti-fibrotic function by a direct interaction with VDR and the retinoid X receptor that promotes their heterodimerization, a process critical for target gene recruitment [163]. Moreover, Duran et al. have shown that loss of p62 expression in HSCs enhances their myofibroblastic differentiation, thereby impairing suppression of fibrosis and inflammation by VDR agonists in chemical agent-induced murine fibrosis and tumor models. Consistent with the aforementioned observations, these findings demonstrate decreased p62 expression to be crucial for myofibroblast differentiation to exert their actions of supporting fibrosis and tumor growth via attenuated VDR signaling. However, the molecular mechanisms underlying the observed down-regulation of p62 expression in activated fibroblasts remain as yet unknown.

Significant down-regulation of p62 expression also underlies the metabolic reprogramming in CAFs mediated by the mammalian target of rapamycin (mTOR) complex 1/Myc cascade controlling IL-6 secretion [166]. Reduced activity of the mTOR complex 1 in p62-deficient fibroblasts accounts for c-Myc down-regulation and the subsequent up-regulation of IL-6, resulting in the promotion of inflammation and tumorigenesis. The lack of p62 in CAFs promotes resistance to glutamine deprivation by directly regulating ATF4 stability via its p62-mediated polyubiquitination [162]. Interestingly, selective autophagy mainly regulated by p62 does not account for the capacity of p62-deficient CAFs to withstand glutamine starvation [162,166]. Up-regulation of ATF4 due to p62 deficiency in the tumor stroma enhances glucose carbon flux through a pyruvate carboxylase-asparagine synthase cascade, which in turn results in asparagine generation as a compensatory source of the nitrogen required for proliferation of both cancer cells and CAFs. It has been shown both in vitro and in vivo that p62-deficient stromal fibroblasts produce non-essential amino acids which are crucial for proliferation in the absence of glutamine by maintaining the TCA cycle in mitochondria, explaining how the p62-deficient tumor stroma stably provides asparagine in an ATF4-dependent manner [162,164,169]. In addition, CSL/RBPJκ, a transcriptional suppressor which is converted into an activator by Notch, plays the role of a negative regulator for CAFs [165]. CSL interacts with p62 and their expression levels are downregulated in murine dermal CAFs through autophagy, indicating that autophagy downmodulates CSL protein expression via p62 in CAFs [165].

## 4. Targeting Tumor Stroma Fibroblasts to Attenuate Tumor Progression

### 4.1. Tumor Stiffness and Enhanced Interstitial Fluid Pressure

As noted above, both tumor cells and CAFs secrete a number of factors which promote angiogenesis, the most widely-accepted of which are members of the VEGF family. Angiogenesis and lymphatic co-options correlate with tumor progression and poor patient outcomes, and are the primary contributors to the altered fluid flow and interstitial fluid pressure (IFP) in the tumor microenvironment [170,171,172]. Vessels developing in the tumor microenvironment are generally irregular and have major gaps in the endothelial cell layer, reducing the degree of coverage by myofibroblasts and pericytes [173,174,175,176]. Furthermore, myofibroblastic CAFs induce not only increases in the numbers of fibrotic foci, but also the contraction of the interstitial space [33,177]. The increased vessel number in conjunction with increased hydraulic conductivity or the relative ease with which fluid moves across the vessel wall, is responsible for the irregular and increased influx of fluid into the tumor stroma. Indeed, rising IFP is frequently reported in solid tumors, such as breast carcinoma, glioblastoma and malignant melanoma [178,179,180]. This increased IFP is due not only to fluid failing to properly drain out of the interstitial space, but also to a number of other physiological changes in the tumor microenvironment, including both an increased number of tumor cells and more ECM deposition in the tumor stroma.

Recent studies support tumor microenvironment stiffness as a therapeutic target aimed at preventing cancer development and progression [181,182,183,184]. The tumor stromal region typically consists of excessive amounts of fibrous collagen, which can be cross-linked by soluble mediators, such as lysyl oxidase (LOX), thereby increasing the stiffness of the tumor microenvironment [185,186,187]. In turn, this increased tumor stiffness is considered to profoundly influence tumor progression inducing activated oncogenic signal pathways driven by activated FAK, Akt, β-catenin, and PI3K, as well as the inhibition of tumor suppressor molecules, such as PTEN. Targeting tumor stiffness via the inhibition of LOX enzymatic activity has been demonstrated to decrease metastatic dissemination of breast and colorectal tumor cells in vivo [188,189]. Treatment with LOX-blocking antibody in combination with gemcitabine also shows attenuated metastases of early-stage pancreatic tumors in Pdx1-Cre KrasG12D/+ Trp53R172H/+ (KPC) mice, however, the effects are not observed in late-stage tumors, presumably due to the presence of considerable levels of already established cross-linked collagen [190].

In pancreatic cancers, tumor cells secrete Hedgehog (Hh) ligands to act on a patched (PTCH1) receptor expressed in CAFs. Hh signaling is thus activated by the ligand binding to PTCH1 that relieves an inhibitory effect on Smoothened (SMO) in these fibroblasts [191]. Hh signaling in CAFs coordinates the acquisition of a poorly-vascularized, desmoplastic microenvironment which impairs drug delivery in pancreatic adenocarcinoma [192,193,194]. Suppression of the stromal Hh signal by the inhibitor of SMO improves the delivery of gemcitabine via transiently increasing vascular density of pancreatic tumors in KPC mice [194].

Increased tumor stiffness impacts not only tumor cells, but also similarly exerts its effects on the surrounding stromal cells, wherein tumor stiffness activates normal fibroblasts to acquire CAF phenotypes and maintains them by the nuclear localization of yes-associated protein (YAP) in the Hippo signal pathway [195,196,197,198]. Actomyosin contractility and Src function are required for YAP activation by stiff matrices. Conversely, YAP depletion reduces the ability of CAFs to form fibrous collagen networks and to promote angiogenesis in vivo. YAP regulates expression levels of several cytoskeleton-related molecules including ANLN and DIAPH3 and then stabilizes MLC2/MYL9. Matrix stiffness further enhances YAP activation, thereby establishing a feed-forward self-reinforcing loop which helps to maintain the CAF phenotype [198]. Increased YAP1 activity in CAFs thus also induces a stiff ECM associated with the Rho-ROCK axis, thereby activating both Src and YAP signaling in a self-stimulating manner.

### 4.2. Therapeutic Strategy Against Activated Tumor Stroma

As noted above, CAFs compromise the effects of cancer therapies not only by producing large amounts of tumor-promoting growth factors and inflammatory cytokines, but also by recruiting other stromal cell types including immunosuppressive inflammatory cells into tumors. Nonetheless, as clonal somatic genetic alterations are rarely harbored in CAFs of different human carcinomas and these fibroblasts are anticipated to be less likely than carcinoma cells to acquire resistance to therapy [199], CAFs are speculated to be a promising therapeutic target [50,200,201].

Treatment with chemotherapy significantly eradicates chemosensitive tumors. However, a considerable number of CAFs often survive in the remnant tumors after treatment. The surviving CAFs acquire innate and adaptive therapeutic resistance that are accompanied by stromal inflammation and increased collagen accumulation, leading to iatrogenic tumor stiffness and the development of chemoresistant tumors. Treatment with several drugs in combination with chemotherapy shows promising results compromising the CAF-induced drug resistance in murine tumor models (Figure 5).

Aberrant IFP elevation disrupts the distribution of systemically administered anti-cancer drugs and thereby compromises the treatment of solid tumors [171,172,202,203]. Hyaluronidases are enzymes that catalyze the degradation of hyaluronic acid (HA), a glycosaminoglycan distributed widely throughout various different tissues. Pegylated recombinant hyaluronidase, known as PEGPH20, contributes to the significant decrease in IFP, thus improving gemcitabine sensitivity in pancreatic ductal adenocarcinoma (PDAC) [204,205,206]. PEGPH20 reportedly reduces IFP in the PDAC microenvironment and expands the tumor vasculature to improve perfusion, which increases access for anti-tumor immune cells and therapeutic agents. A randomized phase II study was also performed using a total of 279 patients with previously untreated metastatic pancreatic ductal adenocarcinoma treated with chemotherapy alone or chemotherapy plus PEGPH20 [207]. The results demonstrated that PEGPH20 treatment is more beneficial in patients with HA-high tumors than in those with HA-low tumors. The level of HA in the tumor-associated stroma was also shown to be a promising biomarker for identifying patients who may benefit from PEGPH20 treatment [207].

Metformin is an oral drug used in the management of patients with type II DM. Metformin administration was first reported to likely be associated with a reduced risk of cancer in the DM patients more than a decade ago [208]. Recently, mounting evidence has pointed to its anti-cancer effects in various malignancies [209,210,211]. This is a typical example of drug re-positioning [212]. Metformin has been reported to play a suppressor role in inflammatory and fibrosis-related diseases, such as atherosclerosis, cardiac fibrosis, renal fibrosis, interstitial pulmonary fibrosis and endometriosis [213,214,215,216]. Inhibition of TGF-β signal pathway, monocyte-to-macrophage differentiation, and NFκB-mediated inflammatory factors is involved in the molecular mechanisms underlying these non-malignant diseases. Importantly, similar mechanisms of metformin action have been suggested to contribute to suppression of the stromal reaction in tumors. In lung cancer, metformin was demonstrated to suppress pulmonary interstitial fibrosis during gefitinib therapy [217]. In ovarian cancer patients, cisplatin administration increases IL-6-producing myofibroblastic CAFs populations through activation of NFκB signal pathway in the tumor-associated stroma [218]. Pretreatment with metformin actually inhibits the desmoplastic stromal reaction via attenuation of the NFκB signal and IL-6 secretion from CAFs. This explains how the IL-6 receptor antagonist, which has conventionally been used for treating rheumatoid arthritis, might serve as an anti-cancer agent in clinical settings.

Renin-angiotensin system inhibitors, which have been prescribed for the treatment of cardiovascular diseases, receive considerable attention in oncology [219]. Angiotensin II (AngII) /AngII type I receptor (AT1R) axis plays pivotal roles in promoting tumor growth and progression. The treatment of CAFs with losartan, which is a selective AT1R blocker (ARB), reportedly attenuates activated fibroblastic state, as exemplified by TGF-β signaling and α-SMA expression, as well as ECM production in culture [220]. Importantly, in mice orthotopically bearing breast cancer cells chemotherapy in combination with losartan inhibits tumor growth and increases survival of the mice more significantly than monotherapy does [219,220,221].

Pirfenidone is an orally active synthetic anti-fibrotic agent structurally similar to pyridine 2,4-dicarboxylate [222]. This drug was recently approved for the treatment of patients with IPF. Pirfenidone exerts anti-fibrotic effects through inhibition of TGF-β and Hh signaling in lung fibroblasts of IPF patients [223]. Miura et al. recently reported that pirfenidone is likely to reduce the risk of lung cancer development in patients with IPF [224]. The retrospective analysis also demonstrated lung cancer incidence to be significantly lower in a pirfenidone-treated group than in a non-pirfenidone-treated group. Indeed, this anti-fibrotic agent induces apoptotic cell death of CAFs residing among NSCLC cells [225] and decreases the expression of collagen triple helix repeat containing 1 (CTHRC1), which is associated with tumor aggressiveness and poor clinical outcomes for NSCLC patients [226]. Importantly, simultaneous co-administration of pirfenidone with chemotherapy inhibits tumor growth and metastasis of breast and pancreatic carcinoma cells, presumably due to attenuation of the TGF-β signal pathway, activated fibroblastic state and ECM protein production in CAFs [227,228,229].

Bromodomain-containing protein 4 (BRD4), a bromodomain and extra-terminal (BET) family member is an important epigenetic reader. BRD4 is critical for the activated fibroblastic state and enhancer-mediated profibrotic gene expression in both HSCs and CAFs [230,231]. Several investigations also show that treatment with a small molecule inhibitor of BRD4 significantly attenuates fibrosis and tumorigenesis via inhibition of stromal TGF-β signaling in murine liver fibrosis models [230,232] as well as patient-derived pancreas and skin squamous cell carcinoma models [231,233].

Experimental evidence supports that the VDR signaling suppresses TGF-β-Smad2/3 signaling to attenuate the fibrotic reactions in fibrosis and tumor stroma [234]. Treatment with VDR ligands thus inhibits the activated state of HSCs and CAFs in murine liver fibrosis [235] and pancreatic carcinoma [236] models. This anti-fibrotic effect is due to the inhibition of the Smad3 recruitment into the binding sites of cis-regulatory regions in the profibrotic genes.

Taken together, these findings indicate that several anti-fibrosis drugs have major potential for impairing and even blocking tumor-promoting CAFs, resulting in attenuation of tumor growth and progression in experimental animal models.

## 5. Closing Remarks

In this review, we have described the close relationship between tumorigenesis and fibrosis, both of which are accompanied by the expansion of activated fibroblast populations. Several aspects of the cellular mechanisms underlying CAF-promoted tumorigenesis and therapy-resistance have been elucidated. However, there are CAF populations that have surprisingly been shown to suppress tumor growth and progression in different murine tumor models including those of the pancreas, bladder and colon [237,238,239,240]. These studies indicate that activation of Hh signaling in CAFs by tumor cell-produced Hh ligand suppresses the growth of tumors via bone morphogenetic protein (BMP) signaling in tumor cells, suggesting the presence of CAF populations with tumor-suppressive functions. In marked contrast, a very recent study using murine models of TNBC found that Hh signaling in CAFs promotes cancer stem cell plasticity and chemoresistance in cancer cells via elevation of stromal FGF5 production [182]. These contradictory observations raise the possibility of cancer cell context-dependent differences in stromal Hh signaling. Although tumor-suppressive or -promoting functions may be inherent in fibroblasts within tumors due to their multiple cells-of-origin, their complex interactions with other stromal cells and carcinoma cells with genetically and epigenetically diverse alterations would also presumably be crucial for generating CAF heterogeneity during tumor progression. Thus, CAFs can reasonably be described as a “cell state” rather than a “cell type” [241].

Several growth factors and cytokines have been identified as inducing CAF differentiation in progenitors, some of which consist of the feedback loop between cancer cells and CAFs in the tumor microenvironment. CAFs also stably maintain their transcriptome and metabolic profiling in an autocrine fashion. It is noteworthy that activated and tumor-promoting traits in these fibroblasts are retained during in vitro propagations, despite a lack of ongoing interactions with carcinoma cells, suggesting the key roles of epigenetic alterations in CAFs, as exemplified by DNA methylation [65]. Given that CAFs are responsible for high IFP in the tumor microenvironment, therapies aimed at preventing iatrogenic tumor stiffness hold great promise. Furthermore, it is remarkable that pirfenidone, one of the anti-fibrotic drugs, not only prevents IPF-associated lung cancer development but also inhibits the distant metastasis of difficult-to-cure breast carcinoma. However, the molecular mechanisms which would allow anti-cancer therapies to precisely target CAFs have yet to be elucidated. The importance of targeting the tumor stroma as well as tumor cells themselves has attracted increasing academic attention as researchers strive to achieve the precision medicine.

## Figures and Tables

**Figure 1 ijms-20-02256-f001:**
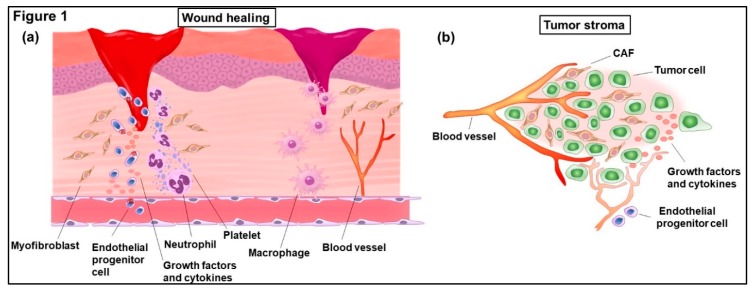
Schematic representation of both wound healing and tumor stroma. Platelets, inflammatory immune cells including neutrophils and macrophages, vascular endothelial cells and activated fibroblasts (myofibroblasts and carcinoma-associated fibroblasts (CAFs)) are recruited into granulation tissues during wound healing (**a**) and tumor stroma (**b**).

**Figure 2 ijms-20-02256-f002:**
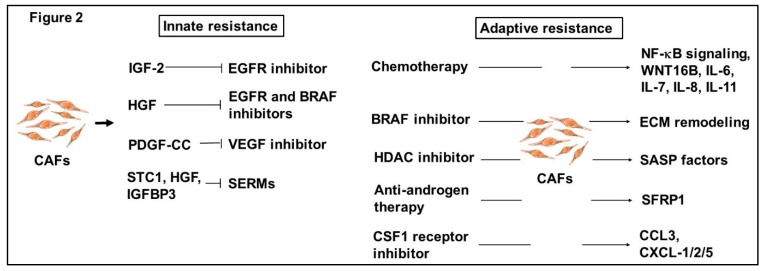
Emerging roles of CAFs for therapeutic resistance. CAFs play crucial roles in innate resistance to anti-cancer drugs (left). CAF-released insulin-like growth factor 2 (IGF2) provides cancer cells with tumor-initiating ability and EGFR-TKI-resistance. HGF produced by CAFs blunts the efficacy of BRAF and EGFR inhibitors in BRAF-mutant melanoma cells and lung cancer cells via MAPK and PI3K/AKT signal pathways. CAF-produced PDGF-CC attenuates the efficacy of anti-VEGF therapy via increasing neo-angiogenesis. Breast tumor cell-derived PDGF-CC also enables CAFs to produce stanniocalcin 1 (STC1), HGF and IGFBP3 that contribute to promoting conversion of luminal cancer cells into basal cancer cells, resulting in resistance to treatment with selective estrogen receptor modulators (SERMs). Upon therapeutic insult, CAFs acquire adaptive resistance (right). Chemotherapy induces pro-inflammatory phenotypes in CAFs via activation of NF-κB signaling, resulting in enhanced production of Wnt family member wingless-type MMTV integration site family member 16B (WNT16B), IL-6 and IL-8 from these fibroblasts that provides breast cancer cells with chemoresistant ability. Increased levels of IL-7 and IL-11 production are also induced in CAFs by chemotherapy, rendering cancer cells tumor-initiating and apoptosis-resistant. Treatment of CAFs with the BRAF inhibitor induces ECM remodeling, resulting in activation of integrin β1/focal adhesion kinase (FAK)/Src and ERK signaling in melanoma cells. The histone deacetylase (HDAC) inhibitor treatment enables CAFs to produce the senescence-associated secretory phenotype (SASP) factors. Exposure to anti-androgen therapy encourages CAFs to produce SFRP1 that promotes prostate cancer neuroendocrine differentiation. Treatment of different human carcinomas with the CSF1 receptor inhibitor targeting TAMs allows CAFs to boost CCL3 and CXCL-1/2/5 productions, resulting in the recruitment of MDSCs into tumors and thus promoting tumor growth and progression.

**Figure 3 ijms-20-02256-f003:**
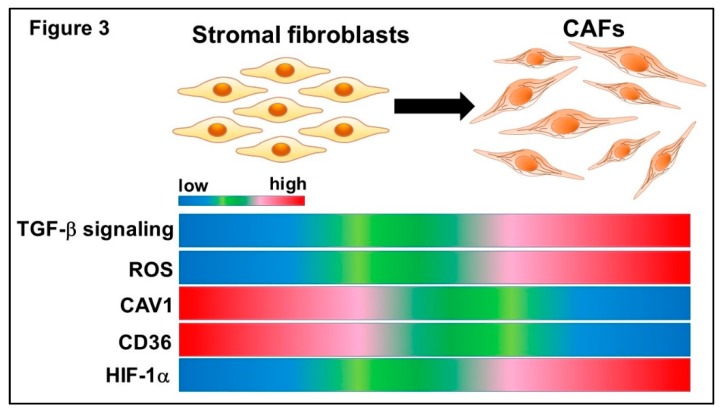
Metabolic reprogramming in CAFs. CD36 and caveolin 1 (CAV1) are components of caveolae, a subset of lipid rafts found in the cell membrane microdomain enriched for cholesterol and signaling proteins. These CD36 and CAV1 expressions are downregulated in CAFs. The attenuated CAV1 expression concomitantly decreases CD36 expression, stabilizes hypoxia-induced factor-1α (HIF-1α), activates TGF-β signal transduction and induces myofibroblast differentiation in fibroblasts. This attenuated CD36 expression also shows a metabolic shift from mitochondrial oxidative phosphorylation to aerobic glycolysis, promoting metabolic plasticity in these fibroblasts. Tumor-derived reactive oxygen species (ROS) are responsible for down-regulation of CAV1 in CAFs. Loss of CAV1 in CAFs also results in ROS elevations, which in turn stabilize HIF-1α.

**Figure 4 ijms-20-02256-f004:**
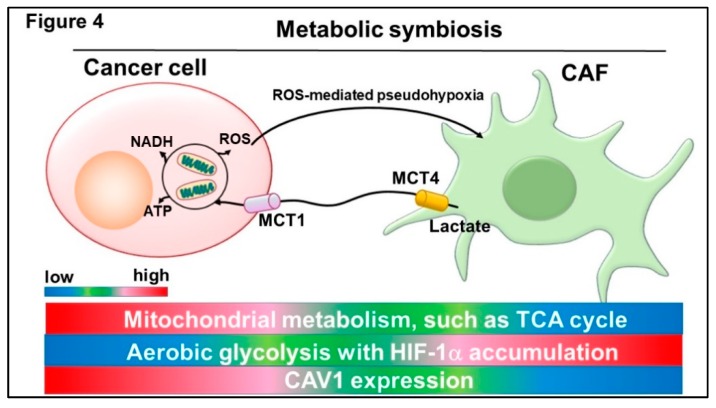
Metabolic symbiosis between cancer cells and CAFs requires the expression of a different MCT subtype. Monocarboxylate transporter 1 (MCT1)-expressing cancer cells induce ROS-mediated pseudohypoxia for MCT4-expressing CAFs, causing HIF-1α accumulation in the nucleus. CAFs depend on aerobic glycolysis and secrete lactate via MCT4. Cancer cells exhibit robust lactate uptake via MCT1, allowing them to generate large amounts of ATP via the mitochondrial TCA cycle. Tumor cells then efficiently produce metabolic intermediates, such as NADH by utilizing lactate derived from CAFs. ROS are a major hallmark of cancer tissues that drives robust metabolism in adjacent proliferating MCT1-positive cancer cells, which are abundant in mitochondria, mediated by the paracrine transfer of mitochondrial fuels, such as lactate, pyruvate and ketone bodies.

**Figure 5 ijms-20-02256-f005:**
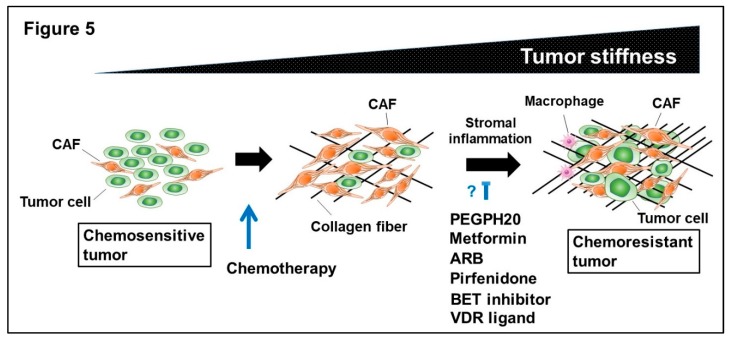
Schematic representation of chemoresistant tumor formation by CAFs and the potential treatment.

**Table 1 ijms-20-02256-t001:** The critical signal pathways activated in both wound-induced fibroblasts and CAFs.

Signal Pathway	Wound-Induced Fibroblasts	CAFs
Epithelial growth factor (EGF)	EGF stimulation increases the phosphorylation of myosin light chain (MLC) subunit of myosin that promotes cell contractility in various different cell types. Activation of PKC with the PKC-δ isoform mediates the cell contraction by EGF-stimulated MLC phosphorylation in murine fibroblast cells [69].	Resistance to the epidermal growth factor receptor (EGFR) tyrosine kinase inhibitor (TKI) is partially medicated by CAFs in tumors through paracrine factors secreted from these fibroblasts [70].
Fibroblast growth factor (FGF)	FGFs have the biological activity of stimulating the proliferation of fibroblasts and angiogenesis [71]. FGFs exert multiple functions through binding to and activation of fibroblast growth factor receptors (FGFRs), and the main signaling through the stimulation of FGFRs is the RAS/MAPK signal pathway.	CAFs secrete increased levels of FGF-1/-3 and promote cancer cell growth and angiogenesis through the activation of FGFR4, which is followed by the activation of extracellular signal-regulated kinase (ERK) and the modulation of MMP-7 expression [72]. In addition, FGF-1 and FGF-3 act as primary autocrine mediators of epithelial-stromal interactions in the tumor progression.
JAK/STAT	Synovial fibroblasts mediate chronic inflammation and joint destruction in patients suffered from rheumatoid arthritis (RA). Increased levels of IL-6, TNF-α and IL-1β production activate STAT3 signaling that in turn boosts expression levels of these cytokines in an autocrine fashion in synovial fibroblasts, promoting chronic inflammation [73]. STAT3 activation also induces receptor activator of nuclear factor kappa B ligand (RANKL) expression that stimulates osteoclastogenesis and thus promotes the joint destruction [73].	CAFs release high levels of IL-6 and CCL2 upon STAT3 activation in co-culture system with cancer cells, promoting the self-renewal and spheroid forming potentials of cancer stem cells [74]. Furthermore, the leukemia inhibitory factor (LIF)-induced JAK1/STAT3 signaling pathway mediates expression of the invasive CAF phenotype [75].
PDGF	PDGFs induce fibroblast activation and fibrosis. PDGF-BB stimulates polarization and provides enhancement and directionality for collagen-driven human dermal fibroblast migration. Akt processes both migratory and proliferative signals from PDGF receptors [76].	Breast tumor cells produce PDGF-CC to activate stromal fibroblasts that in turn confer the basal and estrogen receptor α- negative phenotypes into cancer cells, rendering them unresponsive to endocrine treatment [77].
PGE_2_-Wnt	Dermal fibroblasts expressing a low level of Dickkopf 1, a Wnt signaling antagonist, exhibit enhancement of the canonical Wnt/β-catenin signal pathway with accumulation of prostaglandin E_2_ (PGE_2_) [78]. The PGE_2_ signaling also increases nuclear β-catenin signaling in fibroblasts.	Autocrine activity of PGE_2_ regulates the production of angiogenic factors by fibroblasts, which are key to the vascularization of both primary and metastatic tumor growth [79]. Simultaneous activation of PGE_2_ and Wnt signals in transgenic mice causes gastric cancer with an abundance of vascular endothelial growth factor-A (VEGF-A) expressing CAFs, derived from bone marrow [80].
TGF-β	Upon TGF-β stimulation, fibroblasts are activated and undergo phenotypic transition into myofibroblasts, the key effector cells under fibrotic conditions. The myofibroblast phenotype is characterized by the formation of gap junctions and by the acquisition of a contractile apparatus with associated contractile proteins. In healing wounds, myofibroblasts are required for tissue repair prior to their elimination due to the induction of apoptosis, but constitutively activated myofibroblasts promote fibrosis [81].	Increased TGF-β production by tumor cells gives rise to the desmoplastic stroma in murine tumor models [82,83]. TGF-β potently suppresses immunity, induces angiogenesis and promotes cancer cell migration and invasion by stimulating EMT. Moreover, cancer cell-derived TGF-β activates TGF-β signaling in CAFs, inducing the up-regulation of monocarboxylate transporter 4 (MCT4) (a marker of glycolysis) and BNIP3 (a marker of autophagy) and the loss of caveolin-1 (CAV1) [84].

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
