# Peer review of "Activated Fibroblast Program Orchestrates Tumor Initiation and Progression; Molecular Mechanisms and the Associated Therapeutic Strategies"

_ijms, 2019, doi:10.3390/ijms20092256_

Round 1

Reviewer 1 Report

None

Reviewer 2 Report

agree to accept.

This manuscript is a resubmission of an earlier submission. The following is a list of the peer review reports and author responses from that submission.

Round 1

Reviewer 1 Report

General comments

Authors present a review article describing relationships between tumorigenesis and fibrosis, putting emphasis on the role played by tumor fibroblasts (CAFs) on fibrotic events and tumor progression. I have appreciated that this review article is not written in the conventional (repetitive) way other authors write about CAFs. The focus is more clear on fibrosis, desmaplastic reactions, angiogenesis, tissue stiffness, intersticial fluid pressure and such. An important part of this review is the impact of the mentioned events in therapeutic upshots. Perhaps this point should be better described in title and abstract.

Specific comments

The manuscript is well written and well organized in chapters. Authors aid readers by providing schematic figures to support complex descriptions given in text. However, these “cartoons” are only used in chapter 3 and 4. Authors should consider adding schemes/figures to aid the interpretation of the text also in chapters 1 (“Fibrosis and cancer development”) and 2 (“Fibrosis induced tumor progression”, “Origen and differentiation of CAFs”, “Cross-talk between tumor elements and CAFs”,…).

In chapter 2.2. “Emergence of Treatment-induced CAFs”, authors make an undesirable blend of two different concepts: the pro-tumorigenic phenotype prompted on fibroblasts by stress-induced therapy response, and the CAF role in driving drug resistance. These concepts should be described separately.

In chapter 2.3. “Cross-talk between cancer cells and CAFs”, the chapter title should be changed, as the text refers to interactions of CAFs with different tumor constituents, not only cancer cells (angiogenesis, ECM, desmoplasia, tumor stiffness, IFP, EMT, etc…).

Reviewer 2 Report

Yoshida et al described recent progress and perspective of funtion of fibroblast in the cancer initiation and progression. This review includes various aspects, origin and differentiation into cancer-associated fibroblast (CAFs), cross talk between CAFs and cancer cells, metabolic reprogramming of CAFs during cancer progression, CAFs targeted therapy. Overall the review provides a decent summary of recent progress in the CAFs field. The minor comments:

The review should include more tables or cartoon graph for the readers to easily understand the summary of the text. It would be better the current figures should include a little more information for the readers to understand.

The review can include a section of crosstalk between CAFs and immune cells in the tumor stroma.